# Mitochondrial Dysfunction in Diseases, Longevity, and Treatment Resistance: Tuning Mitochondria Function as a Therapeutic Strategy

**DOI:** 10.3390/genes12091348

**Published:** 2021-08-29

**Authors:** Kazuo Tomita, Yoshikazu Kuwahara, Kento Igarashi, Mehryar Habibi Roudkenar, Amaneh Mohammadi Roushandeh, Akihiro Kurimasa, Tomoaki Sato

**Affiliations:** 1Department of Applied Pharmacology, Graduate School of Medical and Dental Sciences, Kagoshima University, 8-35-1 Sakuragaoka, Kagoshima-City 890-8544, Kagoshima, Japan; y-kuwahara@tohoku-mpu.ac.jp (Y.K.); knt-igrs@dent.kagoshima-u.ac.jp (K.I.); roudkenar@gums.ac.ir (M.H.R.); mohammadi_roushandeh@gums.ac.ir (A.M.R.); tomsato@dent.kagoshima-u.ac.jp (T.S.); 2Division of Radiation Biology and Medicine, Faculty of Medicine, Tohoku Medical and Pharmaceutical University, 1-15-1 Fukumuro, Sendai-City 983-8536, Miyagi, Japan; kurimasa@tohoku-mpu.ac.jp; 3Burn and Regenerative Medicine Research Center, Velayat Hospital, School of Medicine, Guilan University of Medical Sciences, Parastar St., Rasht 41887-94755, Iran

**Keywords:** mitochondria, mitochondrial DNA, clinically relevant radioresistant (CRR) cells, cancer radioresistance

## Abstract

Mitochondria are very important intracellular organelles because they have various functions. They produce ATP, are involved in cell signaling and cell death, and are a major source of reactive oxygen species (ROS). Mitochondria have their own DNA (mtDNA) and mutation of mtDNA or change the mtDNA copy numbers leads to disease, cancer chemo/radioresistance and aging including longevity. In this review, we discuss the mtDNA mutation, mitochondrial disease, longevity, and importance of mitochondrial dysfunction in cancer first. In the later part, we particularly focus on the role in cancer resistance and the mitochondrial condition such as mtDNA copy number, mitochondrial membrane potential, ROS levels, and ATP production. We suggest a therapeutic strategy employing mitochondrial transplantation (mtTP) for treatment-resistant cancer.

## 1. Introduction

A major function of mitochondria is the production of Adenosine tri-phosphate (ATP). Mitochondria use pyruvic acid in the cytoplasm to efficiently produce ATP via the tricarboxylic acid (TCA) cycle and the electron transport chain (ETC). In recent years, it has become clear that mitochondria not only function as an ATP-producing organelle, but also as a signaling center in cell death such as apoptosis and ferroptosis [1,2,3,4]. Mitochondria have also been reported as a major site of reactive oxygen species (ROS) generation [5]. High concentrations of mitochondria-derived ROS are toxic but moderate concentrations have been shown to act as signaling molecules and play an important role in cellular functions, such as cell proliferation [5,6]. Mitochondria-derived ROS also regulate cancer cell growth [6,7,8]. Therefore, mitochondria are very important intracellular organelles that play key roles in normal physiological functions as well as pathophysiological functions. In this review, we discuss and highlight the importance and involvement of mitochondria in normal and disease conditions, specifically focusing on cancer chemoresistance and radioresistance (treatment resistance). We also describe the crucial role of mitochondria in cancer therapy.

### 1.1. Association of Mitochondrial DNA (mtDNA) Mutations in Several Diseases, Longevity, and Radioresistance

Mitochondria have their own DNA (mtDNA). In eukaryotes, almost all the cells except red blood cells have mitochondria and mtDNA, and mtDNA is present in the hundreds to thousands per cell. It has been reported that the mtDNA is inherited maternally [9]. The human mtDNA genome encodes 13 genes, 22 tRNAs, and 2 rRNAs [10]. The 13 genes encoded by mtDNA are all core subunits of oxidative phosphorylation (OXPHOS). OXPHOS is a series of phosphorylation reactions that occur in mitochondria in conjugation with ETC, that is, ATP synthesis reaction (see detail in Section 1.3). It has been reported that mtDNA replication occurs independently of the cell cycle [11]. mtDNA is compacted or relaxed by the concentration of mitochondrial transcription factor (TFAM). When mtDNA is in a relaxed state, mtDNA forms replisome, which is composed of DNA polymerase Gamma (POLγ), mitochondrial DNA helicase (TWINKLE) and mitochondrial single-stranded DNA-binding protein (mtSSB), and replication occur [12,13]. The RNA primers for replication initiation are generated by mitochondrial RNA polymerase (POLRMT) and replication is proceeds. Moreover, mRNA and protein synthesis i.e., transcription and translation in mitochondria are also different from nuclear DNA. The codon of mtDNA is different from nuclear DNA and uses its own tRNA and ribosomal RNA [14]. Unlike nuclear DNA, mtDNA is not protected by histones and is susceptible to gene mutations. The implication of mtDNA mutations in specific diseases, especially mitochondrial diseases has been reported [15,16,17,18]. Mitochondrial disease exhibits various symptoms due to impaired mitochondrial function. Mitochondrial myopathy, encephalopathy, lactic acidosis, and stroke-like episodes (MELAS), myoclonic epilepsy and ragged red fibers (MERRF), chronic progressive external ophthalmoplegia (CPEO), and Leigh syndrome are examples of typical mitochondrial diseases [19]. MELAS is a serious illness with stroke-like symptoms and is diagnosed in early childhood or in the juvenile period [20]. Over 80% of MELAS patients have A3243G mutation and about 10% of MELAS patients have T3271C mutation. These mutations are on the tRNA^Leu(UUR)^. This mutation leads to destabilization of tRNA and leads to reduction of the production of oxidative phosphorylation (OXPHOS) proteins, which produce ATP [21]. MERRF is characterized by myoclonic epilepsy and develops at a relatively old age [22]. About 80% of MERRF patients have A8344G mutation. The symptoms of this disease are reported to be associated with mutations in complexes of NADH-CoQ reductase and cytochrome C-oxidase (COX) [23]. CPEO is characterized by visual muscles’ myopathy and ptosis, pigmentary degeneration of retina, and dysfunction of central nervous system [24]. CPEO is one of the most common mtDNA diseases in adults and caused by point mutation [25] or sporadic large-scale deletions [26,27]. Leigh syndrome is an infantile sub-acute necrotizing encephalopathy. It is a progressive neurodegenerative disease [28]. In this syndrome, it is found that the complex I of ETC is missing [28]. Currently, there is no cure for any mitochondrial disease, and most are treated symptomatically. A number of studies also report that mtDNA mutations have been implicated in other diseases such as deafness [29,30,31], diabetes mellitus [29], Alzheimer’s disease [31], Parkinson’s disease [32], hypertension [33], prostate cancer [34], and exercise intolerance (due to a mutation in the cytochrome b gene) [35]. Figure 1 lists the mtDNA mutations that cause various diseases, as well as mutations that affect longevity or radioresistance [36,37,38,39,40,41,42,43,44,45,46,47].

Several studies have reported that mtDNA mutations play various roles in aging [48,49,50]. Furthermore, it has been shown that various individual mtDNA mutations are present in centenarians [45,46,47]. For example, A5178C mutation, which was found in centenarians, changes the 237th amino acid of ND2 from Leucine to Methionine. Methionine residue in the protein has been reported to have a protective effect on mitochondria against oxidative damage and therefore this mutation is suggested to contribute the longevity at least in part [51]. A5178C mutation has also been reported to protect from myocardial infarction because of the anti-oxidative effect [51]. Moreover, it has been found that there is a mutation (referred to as the “common deletion”) that harbors a 4977 base pair deletion of mtDNA in the D-loop and this deletion increases with age [52]. This deletion has also been implicated in the prognosis of breast cancer [53]. 

A mtDNA mutation (G11778A) has been reported to be implicated in radioresistance [42]. The G11778A mutation is able to repair double strand breaks and leads to short term radiation survival [42]. On the other hand, T8993G mutation has been reported to show radiosensitivity. This mutation is located in the ATP6 gene and decreases ATP synthase, which produce ATP. This mutation has been reported to increase mitochondrial ROS production [44] and show radiosensitivity [54]. 

### 1.2. mtDNA Copy Number and Its Roles in Disease, Longevity, and Treatment Resistance

There are multiple copies of mtDNA in mitochondria. Of note, aged populations have a lower mtDNA copy number in blood [55,56] and exhibit mtDNA heteroplasmy (the presence of more than one type of organelle genome) [57,58]. These results show that mtDNA quantity and quality decrease with age. On the other hand, in the Amami region, which is one of the highest proportions of centenarians, it has been reported that the mtDNA copy number from blood of three groups (under 70, 70–90, over 90 years old) does not decrease with age [59]. In addition, the mtDNA mutation rate did not increase with age. These results indicate that mtDNA quality control plays an important role in longevity.

There are several reports investigating the relationship between mtDNA copy number and cancer. It has been reported that an increase in mtDNA copy number promotes colorectal cancer progression [60]. Additionally, it has been reported that an increase in mtDNA copy number was a risk in breast cancer, pancreatic tumor, lung cancer, lymphomas and skin cancer [61,62,63,64,65,66]. On the other hand, it has been reported that an increase in mtDNA copy number was protective in cancer [67]. In addition, according to findings from the Cancer Genome Atlas project, some cancers have less mtDNA content compared with normal tissue near the tumor [68]. Furthermore, in colorectal cancer, the risk of carcinoma development is associated with lower amount of mtDNA [69]. King et al. [70] established cultured cell lines, which are referred to as “ρ^0^ cells” that lack mtDNA. It is noteworthy that ρ^0^ cells show different behaviors compared to their parental cells. For example, ρ^0^ cells from SK-Hep1 and SH-SY 5Y cells show resistance to oxidative stress [71,72]. In contrast, ρ^0^ cells from yeast and teratocarcinoma cells show sensitivity to oxidative stress [73,74]. We previously showed that ρ^0^ cells are more sensitive to hydrogen peroxide (H_2_O_2_), which is a well-characterized ROS [75]. These findings indicate that there are relationships among mtDNA aging, cancer progression and treatment resistance. However, there is a discrepancy in the relationship between mtDNA depletion and oxidative tolerance. Therefore, we think it is very important to clarify this relationship and to investigate the need for functional mitochondria in cancer cells.

### 1.3. ATP Synthesis, ROS Production, and Mitochondrial Membrane Potential (ΔΨm) in Cancer and Cell Death

Energy production is the main function of mitochondria. Mitochondria produce ATP by OXPHOS. The OXPHOS system is composed of the ETC and ATP synthase. The ETC is composed of complexes I, II, III and IV. ETC transports electrons from complex I to complex IV. During electron transport, a proton gradient is formed over the inner mitochondrial membrane and protons were transported into the mitochondria matrix via ATP synthase. Before the ETC, glucose is metabolized to pyruvate by the glycolysis in the cytosol. Pyruvate then enters into mitochondria by pyruvate dehydrogenase and resulting in mitochondrial acetyl-CoA, nicotinamide adenine nucleotide (NADH)+H, and CO_2_. Acetyl-CoA then enters the tricarboxylic acid (TCA) cycle, which generates further NADH+H. These NADH+H and FADH_2_ from beta-oxidation give an electron to NADH dehydrogenase in complex I and proceed ETC. The ETC system is prone to electron leakage, which generates ROS, i.e., superoxide [76] and H_2_O_2_ [77]. This leakage also induces lipid peroxidation in mitochondrial membranes, which alters the ΔΨm [78,79]. 

It has been reported that cancer cells produce ATP via glycolysis rather than OXPHOS even under aerobic conditions [80,81], which leads to low ΔΨm, resulting in resistance to cell death [82]. It has been proven that ΔΨm is involved in cell death [83]. When the mitochondrial membrane permeable transition pore (mPTP) is opened by a stimulus such as stress, ions and small molecules pass through the membrane and the ΔΨm disappears. 

## 2. Mitochondria Transplantation (mtTP) as a Novel Therapeutic Strategy 

It has been shown that mitochondria can be transferred both artificially and under normal physiological state. We can transfer mitochondria as a “cybrid” [70] or treated isolated mitochondria directly into the cells or tissues [84,85]. We can also transfer mitochondria by co-culture cells as a normal physiological state [86]. Mitochondria transfer from one cell to another cell occurs especially when the mitochondria are injured [87]. Therefore, the mitochondrial transplantation from healthy cells to abnormal cells is thought to be a novel and attractive therapeutic concept. It has been reported that mitochondria and/or organelles transfer between cells through tunneling nanotubes [88]. After the report, replacement of damaged mitochondria with healthy mitochondria has been developed in order to overcome mitochondrial diseases and mitochondria dysfunctions [89,90,91,92,93,94,95,96,97]. It has been shown that mtTP rescues ischemia reperfusion-induced damage and protects the brain from apoptosis [93]. Current clinical and preclinical studies utilizing mtTP have been conducted or are in progress for the treatment of heart ischemia, brain ischemia, sepsis, cancer, acute kidney injury, and theoretically for any disorders in which mitochondria are damaged and disrupted [85,93,98,99]. 

One of the examples of mtTP is mitochondria donation between eggs in fertility treatment. This is a method used in in vitro fertilization called “pronuclear transfer”. This procedure uses a donor egg that has healthy mitochondria. The fertilized donor eggs were enucleated and the nucleus from the mother’s egg, which is also fertilized, was transplanted. This provides the fertilized egg with healthy mitochondria and nuclear DNA from the parent. The embryo is then returned to the mother’s uterus and a healthy baby is born. The United Kingdom passed the first legislation in 2012 to allow the use of mtTP technology on the eggs and fertilized eggs of patients with mitochondrial diseases [100]. In addition, children who have undergone mtTP have already been born [101]. 

We have demonstrated that mitochondria from a non-cancer cell line can be transplanted into cancer cell lines that lack mtDNA (ρ^0^ cells) [94]. This mitochondrial transplantation has been checked using MitoTracker^TM^, which can stain mitochondria, and confirmed that the healthy stained mitochondria from fibroblast cells have certainly transplanted into ρ0 cells. Recently, in a clinical trial, it has been shown that mtTP leads to cardio protection [102]. Moreover, in the breast cancer cell line MCF-7, mtTP induces a decrease in mitochondrial ROS and superoxides via stimulating both superoxide dismutase 2 and catalase expression. Furthermore, mtTP inhibits MCF-7 cell proliferation, reduces cellular oxidative stress, and suppresses drug resistance [103]. It has been reported that mtTP ρ^0^ cells have decreased intracellular Fe^2+^ levels and downregulation of aquaporins. Since aquaporins regulate H_2_O_2_ permeability, these cells exhibit H_2_O_2_ resistance compared with the non-mtTP ρ^0^ cells [96]. Thus, mtTP may enhance mitochondrial function that will allow for the rescue of cells and restoration of normal function. Taken together, these results indicate that mtTP may be an upcoming effective therapeutic option. Therefore, mtTP is a very promising technique, which may be applicable for the treatment of many diseases including cancer. However, mtTP is only in the beginning stages of development, so further investigation will be needed to address various technical and ethical issues. Table 1 shows the preclinical and clinical studies about mitochondrial transplantation.

## 3. Involvement of Mitochondrial Dysfunction in Treatment-Resistant Cancer Cells

In order to investigate the molecular mechanism(s) of radioresistance in cancer cells, we established radioresistant cell lines by step-wise fractionated X-ray exposure [110,111,112,113]. In this procedure, the cells are exposed to X-rays (2 Gy/day) for at least a month, which induces radiation resistance. The established cells were referred to as “clinically relevant radioresistant (CRR)” cells [110]. The morphology of the CRR cells was different from their parental cells and they appeared to be more tightly bound to each other than their parental cells (Figure 2). Moreover, CRR cells exhibit low levels of DNA double strand breaks after ionizing radiation (IR) exposure [110]. In addition, the CRR cells are not only IR resistant but also H_2_O_2_ resistant despite low catalase enzyme activity. Interestingly, the expression of other antioxidative enzyme genes does not seem to be upregulated in CRR cells [114]. CRR cells also exhibit lipid peroxidation resistance upon H_2_O_2_ treatment. Lipid peroxidation normally leads to cell death and this lipid peroxidation resistance was due to a decrease in the expression level of lipoxygenase (ALOX). Administration of oxidized lipids to cancer cells increases cell death and an inhibitor of ALOX decreases lipid peroxidation [114]. Moreover, it has been reported that ALOX targets mitochondria under oxidative stress. For example, when ALOX was administrated into isolated mitochondria, cytochrome c release and ROS generation were observed [115]. Furthermore, it has been reported that ALOX expression was enhanced in CRR cells, and overexpression of ALOX12 enhances ROS generation and amount of HNE, which is one of the lipid peroxidation by-products [116]. These results indicate that the CRR cells inhibit ferroptosis and show resistance from oxidative stress via decreasing mitochondrial function. The characteristics of CRR cells known to date are summarized in Table 2. These results show that plasma membrane status and lipid peroxidation enzyme activity are very important in oxidative stress resistance.

There are additional CRR cell characteristics that may contribute to their treatment resistant phenotype. For example, CRR cells have both low ΔΨm and superoxide production [118]. Furthermore, CRR cells are resistant not only to IR but also to docetaxel, which can increase the level of mitochondrial ROS production [118]. A DNA array experiment showed that CRR cells express higher levels of guanine nucleotide-binding protein 1 (GBP1) compared to parental cells and when GBP1 is knocked down by siRNA, CRR cells lose their radioresistance [120]. Recently, it has been reported that knockdown of GBP1 results in impaired mitochondrial respiratory function [121]. Treatment with everolimus, an mTOR inhibitor, abolishes the IR resistance properties of CRR cells [110]. In addition, the autophagy inducer rapamycin increases the radiosensitivity of CRR cells and the autophagy inhibitor 3-methyladenine induces radioresistance in parental cells [122]. Furthermore, an mTOR inhibitor affects mitochondria dynamics [123]. These results show strong relationships between radioresistance, autophagy, and mitochondria. There is also a correlation between radioresistance and mtDNA copy number. For example, mtDNA copy number was decreased in CRR cells compared to parental cells [112]. Furthermore, CRR cells had low ATP production, low ROS levels, low ΔΨm, and low aquaporin 8 gene expression, of which the latter is expressed in both the plasma and mitochondrial membranes [113,114,124]. MicroRNA array analysis revealed that CRR cells had higher miR-7-5p expression levels compared to parental cells [119]. Candidate target genes of miR-7-5p are summarized in Table 3. One of the target genes is SLC25A37 (mitoferrin), an iron transporter in mitochondria. When this gene was knocked down by siRNA, radioresistance was observed in parental cells [119]. Moreover, mitochondrial Fe^2+^ levels were significantly decreased in CRR cells [119]. Mitoferrin is a mitochondrial iron importer that synthesizes mitochondrial heme and iron–sulfur clusters. These results suggest that mitoferrin have an important role in CRR cell characteristics. Recently, inhibition of mir-7-5p decreased intracellular and mitochondrial ROS, enhanced JC-1 signal, which is an indicator of ΔΨm, downregulated the ferritin gene expression, and enhanced the ALOX12 gene expression [116]. In contrast, ρ^0^ cells show high Fe^2+^ amount, high lipid peroxidation, and low ALOX expression. These factors are different (opposite) from CRR cells and ρ^0^ cells considered to be sensitive to the oxidative stress. Interestingly, CRR cells lose their radioresistance when irradiation is terminated, and the cells are cultured for more than six months [119]. This result strongly suggests that this phenotype is reversible and radioresistance induced by irradiation also has the potential for reversibility. Therefore, further investigation of CRR cells is very important to eradicate cancer. Recently, it has been reported that an Italian group has established a cell line in rhabdomyosarcoma, also named clinically relevant radioresistant cells [125]. They establish these cells by irradiating 6 Gy × 6 times not 2 Gy/day, but show radioresistance. These cells have been reported to produce less mitochondrial superoxide. Taken together, these data show that mitochondria play key roles in cancer therapy and resistance to treatment.

## 4. Conclusions and Future Perspectives

mtDNA mutations and mtDNA copy number are important not only for health, mitochondrial diseases, and aging but also for cancer radioresistance. Other mitochondrial parameters such as ATP production, ΔΨm, and ROS production are also involved in radioresistance. mtTP is now ready for clinical evaluation and this technology may be a promising therapeutic strategy for a variety of diseases with mitochondrial dysfunctions such as mitochondria diseases, myocardial infarction, acute kidney injury, aging, and cancer. To apply this technology, further investigation addressing various ethical and technical issues will be required. Additionally, a better understanding of the underlying mechanism of cancer cell resistance particularly clarifying the role of mitochondria in this process would lead to the development of more effective therapeutic strategies for cancer. Overall, in cancer cells it may be possible to fine-tune mitochondria function so that radioresistance might be overcome (summary in Figure 3). However, further investigation including animal studies and clinical trials are required in order to determine if altering mitochondrial function can confer radiosensitivity to cancer cells. 

## Figures and Tables

**Figure 1 genes-12-01348-f001:**
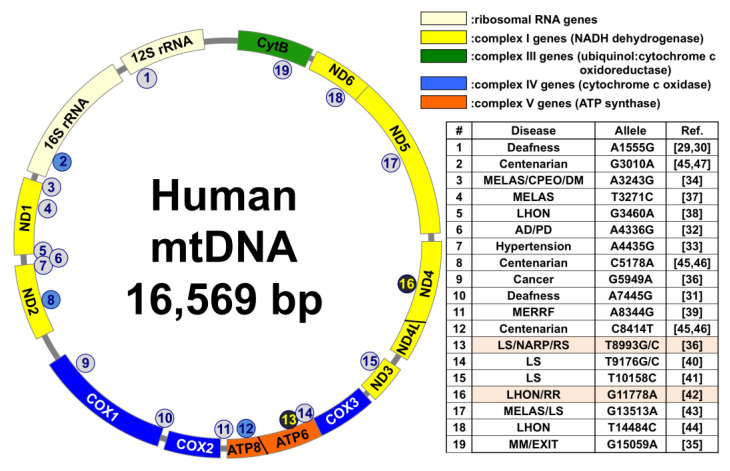
Mitochondrial DNA (mtDNA) mutations that cause various diseases, as well as mtDNA mutations that affect longevity or radioresistance. Centenarian: a person over 100 years old; MELAS: mitochondrial myopathy, encephalopathy, lactic acidosis, and stroke-like episodes; CPEO: chronic progressive external ophthalmoplegia; DM: diabetes mellitus; LHON: Leber’s hereditary optic neuropathy; AD/PD: Alzheimer’s and Parkinson’s diseases; MERRF: myoclonic epilepsy and ragged red fibers; LS: Leigh syndrome; NARP: neuropathy, ataxia, and retinitis pigmentosa; RS: radiosensitive; RR: radioresistance; MM: mitochondrial myopathies; EXIT: exercise intolerance.

**Figure 2 genes-12-01348-f002:**
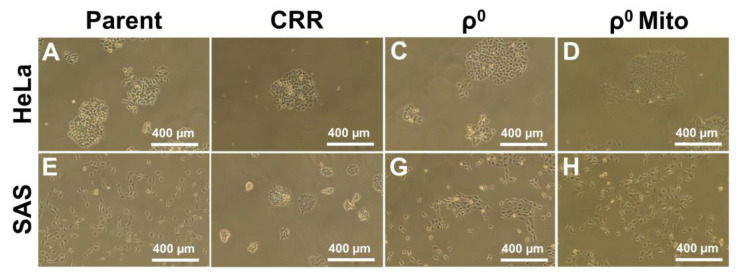
Morphology of CRR cells, ρ^0^ cells, ρ^0^ cells harboring transferred mitochondria, and parental cells. (**A**): HeLa parent cells, (**B**): HeLa CRR cells, (**C**): HeLa ρ^0^ cells, (**D**): HeLa ρ^0^ Mito cells, (**E**): SAS parent cells, (**F**): SAS CRR cells, (**G**): SAS ρ^0^ cells, (**H**): SAS ρ^0^ Mito cells.

**Figure 3 genes-12-01348-f003:**
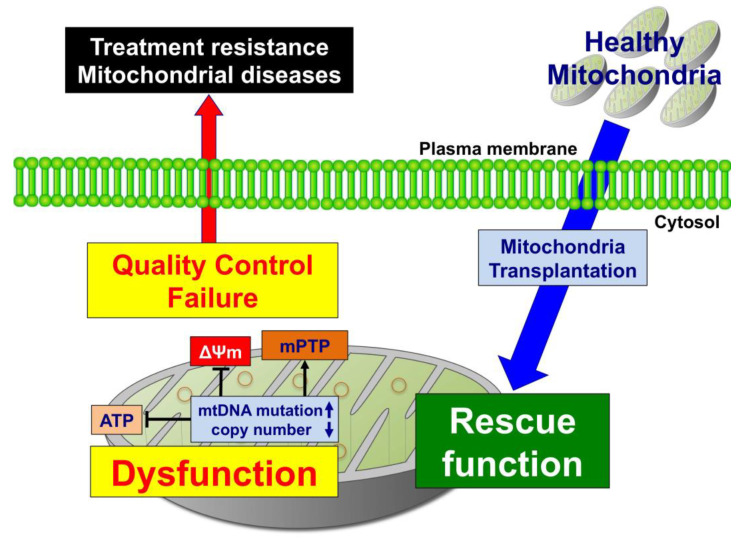
Mitochondria transplantation (mtTP) is a candidate strategy designed to rescue mitochondrial quality control failure. Mitochondrial dysfunction induced by mtDNA mutation or a decrease in mtDNA copy number leads to a decrease in ATP production, a decrease in mitochondrial membrane potential (ΔΨm), and opening of the mitochondrial membrane permeable transition pore (mPTP). This mitochondrial quality control failure induces treatment resistance and mitochondrial disease. Mitochondrial quality control failure may be rescued by transplantation of healthy mitochondria.

**Table 1 genes-12-01348-t001:** Preclinical and clinical studies about mitochondrial transplantation.

Donor	Recipient	Disease	Result	Reference
Rectus Abdominis	Heart	Heart ischemia reperfusion	Cardiac function improved	[85,98]
Granular cells	Oocyte	Infertility	Normal babies were born	[101]
Astrocytes	Neuron	Ischemic damage	Recover ATP production	[104]
HeLa cells (cervical cancer cell line)	AD model mice	Alzheimer disease	Cognitive defect and gliosis were ameliorated	[105]
Cybrids from PC-12 cells and human osteosarcoma	Brain	6-OHDA induced PD model	Improve motor function and mitochondrial function	[106]
BHK-21 cell (kidney cell line)	Sciatic nerve	Sciatic nerve crush	Injured sciatic nerve improved	[107]
Oocyte cytoplasm	Oocyte	Infertility	Increase pregnancy	[108]
Mesenchymal stem cells	Brain	Rat brain ischemia reperfusion	Protect from apoptosis Restores motor function	[94]
WI-38 (fibroblast cell line)	ρ^0^ cells (HeLa, SAS)	mtDNA deficient	Prohibitin 2 enhancement Survive without pyruvate and uridine	[97]
MLO-Y4 cell (osteocyte cell line)	ρ^0^ cells (MLO-Y4)	mtDNA deficient	Increase ATP production	[109]

**Table 2 genes-12-01348-t002:** Characteristics of clinically relevant radioresistant (CRR) cells.

CRR Characteristics		References
Morphology	Tight binding	This review, [116]
Irradiation	Resistant	[111,117]
H_2_O_2_	Resistant	[114]
Docetaxel	Resistant	[118]
DNA DSB	Low	[112]
ΔΨm	Low	[118]
Superoxide	Low	[114]
Hydroxyl radical	Low	[114]
Lipid peroxidation	Low	[114]
mtDNA copy number	Low	[114]
ATP production	Low	[114]
Fe^2+^ amount	Low	[119]
AQP8 gene expression	Low	[114]
ALOX gene expression	Low	[114]
GBP1 gene expression	High	[120]
miR-7-5p expression	High	[119]

**Table 3 genes-12-01348-t003:** miR-7-5p target genes.

Localization	Gene Name		
Plasma membrane	ATP2B2	FLRT2	SEMA4C
	SEAMA6D	TMEM65	VSTM4
Cytoplasm	AKT3	MAPK4	-
Mitochondria	CRLS1	NDFUA4	PTPMT1
	SLC25A15	SLC25A16	SLC25A37
	TIMM50	TMEM65	VDAC1
ER	SERP1	-	-
Lysosome	BLOC1S4	-	-
Golgi apparatus	GLG1	GOLGB1	-

ATP2B2: ATPase plasma membrane Ca^2+^ transporting 2; FLRT2: Fibronectin Leucine Rich Transmembrane Protein 2; SEMA4C: Semaphorin 4C; SEAMA6D: Semaphorin 6D; TMEM65: Transmembrane Protein 65; VSTM4: V-Set And Transmembrane Domain Containing 4; AKT3: AKT Serine/Threonine Kinase 3; MAPK4: Mitogen-Activated Protein Kinase 4; CRLS1: Cardiolipin Synthase 1; NDFUA4: NADH dehydrogenase (ubiquinone) 1 Alpha subcomplex subunit 4; PTPMT1: Protein Tyrosine Phosphatase Mitochondrial 1; SLC25A15: Mitochondrial ornithine transporter 1; SLC25A16: Graves disease carrier protein; SLC25A37: Mitoferrin-1; TIMM50: Translocase Of Inner Mitochondrial Membrane 50; TMEM65: Transmembrane protein 65; VDAC1: Voltage-dependent anion-selective channel protein 1; SERP1: Stress Associated Endoplasmic Reticulum Protein 1; BLOC1S4: Biogenesis Of Lysosomal Organelles Complex 1 Subunit 4; GLG1: Golgi Glycoprotein 1; GOLGB1: Golgin B1.

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
