# Peer review of "Mitochondrial Dysfunction in Diseases, Longevity, and Treatment Resistance: Tuning Mitochondria Function as a Therapeutic Strategy"

_genes, 2021, doi:10.3390/genes12091348_

Round 1

Reviewer 1 Report

Summary

This is purported to be a review of mitochondrial dysfunction in disease, longevity and treatment resistance, and of mitochondrial transfer as a novel therapeutic strategy. However, while some sections are better than others, on the whole it falls short of adequately reviewing the current literature. 

General

1.. In a review commenting on mtDNA genomic aberrancies I would have expected not only a breadth of mitochondrial functions to be covered but also a more detailed explanation of the structure and function of mtDNA, also the OXPHOS system and the relationship between the two. I would also have expected a discussion on how mtDNA copy number is regulated, and how aberrations in terms of mutations or different levels of mtDNA can lead to issues in the OXPHOS and related processes - I do not feel this has been covered in enough detail. This would be best placed in the introduction

  1. I would also have expected throughout this review, with perhaps the exception of section 3 on treatment resistant cells, a greater breadth of literature to have been covered by the review to add value to the field, and also some information on the approach including selection criteria as to why certain literature and not others was included in the review. What is presented here e.g. on mtDNA mutations very much feels like a snap-shot of a very restricted range of literature, and also includes very limited (almost no) detail and critique of the cited papers is given. Things to consider are which mutations exactly cause these diseases, changes in longevity, and treatment sensitivity? How were they discovered, i.e. what were the methods used and what was the study design used in each paper? It's very clear that there are many mutations in mtDNA compared to reference sequence, but what evidence do the papers present to show they really are the causal factor/ or strongly associated with the disease, longevity and/or treatment resistance? What cohorts, including size, were investigated, so that we can interpret the findings accordingly? What are the mechanisms by which the mutations are thought to contribute to the disease, changes in longevity and/ or treatment sensitivity?

Detailed

Title:

Line 3: Mito transfer between cells is not novel and dates back to when the cybrids were made by King and Attardi

Abstract:

Line 20: I dont see how mutation in mtDNA can lead to longevity, surely you mean altered lifespans?

Line 21: “Moreover,…aging” - Therefore by implication differences in longevity? MtDNA copy no differences are also linked to disease and differences in treatment sensitivity so could you not alter the sentence before to include mtDNA mutation and copy no differences?

Lines 21-22: “In addition,…cell death” - I am not sure this sentence adds anything that has not already been said - I suggest deleting.

Lines 22-23: - “The…cancer therapy” - Similarly I dont think this sentence adds much, I feel you are trying to comment on the mechanism by which mtDNA mutation and copy number might influence disease, longevity and treatment sensitivity, but it does not really come across/ achieve this.

Lines 23-26: “In this review,…cancer resistance” - If you only focus on this why then do you mention other disease and longevity in the title and earlier in the abstract - I feel you should either change the title or this part, so that the two align better, depending indeed on the content of the review.

Section 1: Intro:

Line 32: Re “ATP” - All abbreviations in full at first mention, then use the abbreviation thereafter - please check the remainder of the manuscript for other corrections like this that you need to make.

Line 36: “also” should be “but also” and “deaths” should be “death” - Given this is a review, I would have expected a greater range of mitochondrial functions to have been discussed, not just a basic few.

Line 41: “…and cell death” - Surely if the levels are sufficient to cause death then they are toxic - I would delete cell death in this instance.

Line 47: You only mention radio resistance in the abstract, please also mention chemo resistance.

Lines 50-87: I would have expected a greater breadth of literature to have been covered for a review to add value to the field, and also some information on the approach i.e. including selection criteria as to why certain literature and not others was included in the review. What is presented here very much feels like a snap-shot of a very restricted range of literature, and also includes very limited (almost no) detail and critique of the cited papers is given. Things to consider are which mutations exactly cause these diseases, changes in longevity, and treatment sensitivity? How were they discovered, ie what were the methods used and what was the study design used in each paper? It's very clear that there are many mutations in mtDNA compared to reference sequence, but what evidence do the papers present to show they really are the causal factor/ or strongly associated with the disease, longevity and/or treatment resistance? What cohorts, including size, were investigated, so that we can interpret the findings accordingly? What are the mechanisms by which the mutations are thought to contribute to the disease, changes in longevity and/ or treatment sensitivity?

Line 80: Exclusively in centenarians?

Line 83: “D-loop” not “d-loop”.

Lines 96-97: The way the findings have been articulated here, I do not think that they do show a relationship between mtDNA copy no and aging. Again, I think it’s because there is not enough detail and explanation provided by the authors. Do you mean if we look at the general population (that might have a lower mean lifespan) mtDNA copy no goes down over time? While if you look at specific populations such as in the Amani region (that have a higher mean lifespan due to more centenarians) mtDNA copy no remains stable and could therefore explain the enhanced longevity? If so, I think this could be much better expressed, and also a comment on the mechanism by which differences in mtDNA copy no could influence longevity should be included.

Line 99: Re ref [44] - This is just one study! I know for a fact there are several studies looking at the relationship between mtDNA copy no and different types of cancer, and the relationship is not clear cut. Some studies show the opposite to this. A greater breadth of literature should be considered/ critiqued for a more systematic and balanced review that's also representative of the current literature in the field.

Line 106-107:  This is a huge unjustified statement, what you have covered does not indicate this.

Line 110: I think you mean “Energy production”?

Line 111: You need to say the OXPHOS system is comprised of the ETC and ATP synthase. Say what the ETC is. See earlier general comment on including more detail on the OXPHOS system.

Lines 117-118: Caution with the use of the word "reported" here and elsewhere. In several cases it really is the excepted dogma that x is involved with y, so in this case membrane potential a lowering of it can elicit apopotosis, so it would be more appropriate (as it has been shown countless times) that changes is mitochondrial membrane potential can cause cell death. It's not just reported, by now the evidence really suggests it is fact and so should be written as such.

Section 2: mtTP:

Line 123: Do you mean mitochondria can be transferred artifically or under normal physiological circumstances, be more explicit because there are studies on both.

Line 123: “…are injured.” – missing references.

Lines 125-126: Be far more explicit that you mean the transfer if mitochondria from healthy to abnormal cells.

Lines 139-142: It seems out of place to start talking about mitochondrial donation between eggs/embryos in terms of the prevention of inherited mtDNA disease with no introduction up until this point. I think if you are using mito donation as a justfication for treating other diseases eg cancer then there should be more background included on it before this point, and even then the link is somewhat tenious as introducing mtDNAs into eggs is rather different than between healthy and abnormal somatic cells.

Lines 142-150: I think the flow of this section could be better:

-talk about mito donation between eggs as an example of mtTP

-then talk about mito donation between healthy cell to a rho-0 as a proof of principle for non-germ/ embryonic cell mito transfers

-then talk about other mito transfers between other cell types, otherwise it is very hard to follow to go from one to the other and then back again.

Section 3: Treatment resistant cells:

Lines 159-230: This section is far more comprehensive; the other sections need to be brought up to this level.

Tables and figures:

Figure 1: The formatting of the wording is off.

Table 1: Once more this is a summary of an incredibly limited review of the available literature that is out there on mitochondrial transfer. This should be addressed.

Author Response

Reviewer 1

This is purported to be a review of mitochondrial dysfunction in disease, longevity and treatment resistance, and of mitochondrial transfer as a novel therapeutic strategy. However, while some sections are better than others, on the whole it falls short of adequately reviewing the current literature.

General

1.. In a review commenting on mtDNA genomic aberrancies I would have expected not only a breadth of mitochondrial functions to be covered but also a more detailed explanation of the structure and function of mtDNA, also the OXPHOS system and the relationship between the two. I would also have expected a discussion on how mtDNA copy number is regulated, and how aberrations in terms of mutations or different levels of mtDNA can lead to issues in the OXPHOS and related processes - I do not feel this has been covered in enough detail. This would be best placed in the introduction

We added the detailed explanation of the function of mtDNA and OXPHOS system and the relationship between mtDNA and OXPHOS in the introduction section.

I would also have expected throughout this review, with perhaps the exception of section 3 on treatment resistant cells, a greater breadth of literature to have been covered by the review to add value to the field, and also some information on the approach including selection criteria as to why certain literature and not others was included in the review. What is presented here e.g. on mtDNA mutations very much feels like a snap-shot of a very restricted range of literature, and also includes very limited (almost no) detail and critique of the cited papers is given. Things to consider are which mutations exactly cause these diseases, changes in longevity, and treatment sensitivity? How were they discovered, i.e. what were the methods used and what was the study design used in each paper? It's very clear that there are many mutations in mtDNA compared to reference sequence, but what evidence do the papers present to show they really are the causal factor/ or strongly associated with the disease, longevity and/or treatment resistance? What cohorts, including size, were investigated, so that we can interpret the findings accordingly? What are the mechanisms by which the mutations are thought to contribute to the disease, changes in longevity and/ or treatment sensitivity?

We added the detailed explanation for each mutation and try to explain the mechanism of the phenotype that is represented by the mutation. Please check each answer to the comment we submitted.

Detailed

Title:

Line 3: Mito transfer between cells is not novel and dates back to when the cybrids were made by King and Attardi

We removed “novel emerging” from the title.

Abstract:

Line 20: I dont see how mutation in mtDNA can lead to longevity, surely you mean altered lifespans?

We think that the certain mutation of mtDNA lead to longevity. For example, A5178C mutation, which was found in centenarians, changes the 237th amino acid of ND2 from Leucine to Methionine. Methionine residue in the protein has been reported to have a protective effect on mitochondria against oxidative damage and therefore this mutation is suggested to contribute the longevity at least in part. However, no explanation was described in the previous manuscript. Therefore, we added the explanation and references about the longevity and mtDNA mutation where centenarian first appears in the text.

(In lines 97-104 of page 3)

Line 21: “Moreover,…aging” - Therefore by implication differences in longevity? MtDNA copy no differences are also linked to disease and differences in treatment sensitivity so could you not alter the sentence before to include mtDNA mutation and copy no differences?

We combined the two sentences into one and made “Mitochondria have their own DNA (mtDNA) and mutation of mtDNA or change the mtDNA copy numbers leads to disease, cancer radioresistance and aging including longevity.”

(In lines 19-21 of page 1)

Lines 21-22: “In addition,…cell death” - I am not sure this sentence adds anything that has not already been said - I suggest deleting.

We delete this sentence.

Lines 22-23: - “The…cancer therapy” - Similarly I dont think this sentence adds much, I feel you are trying to comment on the mechanism by which mtDNA mutation and copy number might influence disease, longevity and treatment sensitivity, but it does not really come across/ achieve this.

We delete this sentence.

Lines 23-26: “In this review,…cancer resistance” - If you only focus on this why then do you mention other disease and longevity in the title and earlier in the abstract - I feel you should either change the title or this part, so that the two align better, depending indeed on the content of the review.

We have mentioned various disase, longevity, and cancer resistance, each of which is included in the text, to match the title and content. We correctred “In this review, we discuss the mtDNA mutation, mitochondrial disease, longevity, and importance of mitochondrial dysfunction in cancer first. In the later part, we particularly focus on the role in cancer resistance and the mitochondrial condition such as mtDNA copy number, mitochondrial membrane potential, ROS levels, and ATP production. mitochondrial membrane potential, ROS levels, and ATP production in cancer resistance.” in this part.

(In lines 21-25 of page 1)

Section 1: Intro:

Line 32: Re “ATP” - All abbreviations in full at first mention, then use the abbreviation thereafter - please check the remainder of the manuscript for other corrections like this that you need to make.

For abbreviations such as ATP, the official name (abbreviation) was described when it first appeared, and the abbreviations are described thereafter.

(For example, In lines 31-32 of page 1)

Line 36: “also” should be “but also” and “deaths” should be “death” - Given this is a review, I would have expected a greater range of mitochondrial functions to have been discussed, not just a basic few.

We changed “also” to “but also” and “deaths” to “death”.

(In lines 35-36 of page 1)
We try to describe other mitochondrial functions in other section with references.

Line 41: “…and cell death” - Surely if the levels are sufficient to cause death then they are toxic - I would delete cell death in this instance.

We delete “and cell death”.

Line 47: You only mention radio resistance in the abstract, please also mention chemo resistance.

We described “chemo/radioresistance” in the abstract.

(In lines 20 of page 1)

Lines 50-87: I would have expected a greater breadth of literature to have been covered for a review to add value to the field, and also some information on the approach i.e. including selection criteria as to why certain literature and not others was included in the review. What is presented here very much feels like a snap-shot of a very restricted range of literature, and also includes very limited (almost no) detail and critique of the cited papers is given. Things to consider are which mutations exactly cause these diseases, changes in longevity, and treatment sensitivity? How were they discovered, ie what were the methods used and what was the study design used in each paper? It's very clear that there are many mutations in mtDNA compared to reference sequence, but what evidence do the papers present to show they really are the causal factor/ or strongly associated with the disease, longevity and/or treatment resistance? What cohorts, including size, were investigated, so that we can interpret the findings accordingly? What are the mechanisms by which the mutations are thought to contribute to the disease, changes in longevity and/ or treatment sensitivity?

We have added the references and mentioned the factors that cause disease and lifespan changes with certain mtDNA mutations.

(In lines 61-77 of page 2)

Line 80: Exclusively in centenarians?

A5178C mutation has been reported not only to be involved in longevity but also reported to be protective against myocardial infarction by anti-oxidative effect. We mentioned in the text.

(In lines 102-103 of page 3)

Line 83: “D-loop” not “d-loop”.

We corrected, “d-loop” to ”D-loop”.

Lines 96-97: The way the findings have been articulated here, I do not think that they do show a relationship between mtDNA copy no and aging. Again, I think it’s because there is not enough detail and explanation provided by the authors. Do you mean if we look at the general population (that might have a lower mean lifespan) mtDNA copy no goes down over time? While if you look at specific populations such as in the Amani region (that have a higher mean lifespan due to more centenarians) mtDNA copy no remains stable and could therefore explain the enhanced longevity? If so, I think this could be much better expressed, and also a comment on the mechanism by which differences in mtDNA copy no could influence longevity should be included.

This section describes the decrease of mtDNA copy number and increase of mtDNA mutation with aging. And in the Amami area, where there are many centenarians, the number of mtDNA copies and the rate of mutations have not decreased with age. Therefore, we mentioned above and described that these results indicate that mtDNA quality control plays an important role in longevity.

(In lines 124-127 of page 4)

Line 99: Re ref [44] - This is just one study! I know for a fact there are several studies looking at the relationship between mtDNA copy no and different types of cancer, and the relationship is not clear cut. Some studies show the opposite to this. A greater breadth of literature should be considered/ critiqued for a more systematic and balanced review that's also representative of the current literature in the field.

We added references other than 44, including the opposite results, and described that they were related to cancer progression and treatment resistance, although no conclusions were reached.

(In lines 128-138 of page 4)

Line 106-107: This is a huge unjustified statement, what you have covered does not indicate this.

We described, “These findings indicate that there are relationships among mtDNA aging, cancer progression and treatment resistance.”

(In lines 145-147 of page 4)

Line 110: I think you mean “Energy production”?

Yes. We changed the “Energy Production” in the text.

(In line 150 of page 4)

Line 111: You need to say the OXPHOS system is comprised of the ETC and ATP synthase. Say what the ETC is. See earlier general comment on including more detail on the OXPHOS system.

We mentioned that the OXPHOS system is composed of the ETC and ATP synthase. We also explained about ETC in the text.

(In lines 151-162 of pages 7-8)

Lines 117-118: Caution with the use of the word "reported" here and elsewhere. In several cases it really is the excepted dogma that x is involved with y, so in this case membrane potential a lowering of it can elicit apopotosis, so it would be more appropriate (as it has been shown countless times) that changes is mitochondrial membrane potential can cause cell death. It's not just reported, by now the evidence really suggests it is fact and so should be written as such.

We changed “reported” to ”proved” in this sentence.

(In line 168 of page 5)

Section 2: mtTP:

Line 123: Do you mean mitochondria can be transferred artifically or under normal physiological circumstances, be more explicit because there are studies on both.

We described both of the transfer and explain them in the text. We think that the transplantation of mitochondria by artificially will be a new therapeutic strategy for cancer treatment especially for treatment-resistant cancer.

Line 123: “…are injured.” – missing references.

We add the reference.

(In line 180 of page 5)

Lines 125-126: Be far more explicit that you mean the transfer if mitochondria from healthy to abnormal cells.

We add “healthy cells to abnormal cells” in the text.

(In lines 180-181 of page 5)

Lines 139-142: It seems out of place to start talking about mitochondrial donation between eggs/embryos in terms of the prevention of inherited mtDNA disease with no introduction up until this point. I think if you are using mito donation as a justfication for treating other diseases eg cancer then there should be more background included on it before this point, and even then the link is somewhat tenious as introducing mtDNAs into eggs is rather different than between healthy and abnormal somatic cells.

We delete about mitochondrial donation here and moved to the next section. The mitochondrial donation is described in the next paragraph after the line break.

(In lines 191 of page 5)

Lines 142-150: I think the flow of this section could be better:

-talk about mito donation between eggs as an example of mtTP

-then talk about mito donation between healthy cell to a rho-0 as a proof of principle for non-germ/ embryonic cell mito transfers

-then talk about other mito transfers between other cell types, otherwise it is very hard to follow to go from one to the other and then back again.

Thank you for your suggestion. We rewrite this section as you recommended.

Section 3: Treatment resistant cells:

Lines 159-230: This section is far more comprehensive; the other sections need to be brought up to this level.

Thank you for your comments. Based on your comments, we've added content to other sections to bring it closer to the same level as this section. Thanks to you, this review became a better one.

Tables and figures:

Figure 1: The formatting of the wording is off.

We correct some wording format.
LHON: Leber’s hereditary optic neuropathy.
NARP: Neuropathy, ataxia, and retinitis pigmentosa.

Table 1: Once more this is a summary of an incredibly limited review of the available literature that is out there on mitochondrial transfer. This should be addressed.

We added references about mitochondrial transplantation and we remade the Table 1.

Reviewer 2 Report

This manuscript entitled “Mitochondrial dysfunction in disease, longevity, and treatment resistance: tuning mitochondria function as a novel emerging therapeutic strategy” by Tomita T et al. is a short review covering the relationship between mitochondrial function, mitochondrial DNA and cell death related to diseases.  I have a couple of inquiries.

  1. In lines 103-107 of page 3, several papers have been reported resistance of rho0 cells against cell death. Such papers need to be referred and discussed regarding the mechanisms of different behavior.

  2. In lines 170-180 of page 5, cell death by peroxidized lipids is related to ferroptosis. Please discuss the relationship between mitochondrial function of CRR cells and ferroptosis.

  3. In lines 214-219 of page 7, please discuss the significance regarding mitoferrin.

  4. I cannot find Fig. 3.

  5. The authors have reported that rho0 cells are more prone to ferroptosis. Please discuss such findings in comparison with the findings in CRR cells.  

  6. In general, total content of the manuscript is relatively small as a review. If possible, expansion of the content is recommended.

Author Response

Reviewer 2

This manuscript entitled “Mitochondrial dysfunction in disease, longevity, and treatment resistance: tuning mitochondria function as a novel emerging therapeutic strategy” by Tomita T et al. is a short review covering the relationship between mitochondrial function, mitochondrial DNA and cell death related to diseases. I have a couple of inquiries.

In lines 103-107 of page 3, several papers have been reported resistance of rho0 cells against cell death. Such papers need to be referred and discussed regarding the mechanisms of different behavior.

We added references that the ρ0 cells show resistance against cell death. We describe “There is a discrepancy in the relationship between mtDNA depletion and oxidative tolerance. Therefore, we think it is very important to clarify this relationship and to investigate the need for functional mitochondria in cancer cells.” in the text.

(In lines 141-147 of page 4).

In lines 170-180 of page 5, cell death by peroxidized lipids is related to ferroptosis. Please discuss the relationship between mitochondrial function of CRR cells and ferroptosis.

We described the discussion about the relationships between mitochondrial function of CRR cells and ferroptosis.

(In lines 253-257 of page 7)

In lines 214-219 of page 7, please discuss the significance regarding mitoferrin.

We discuss the significance regarding mitoferrin in the text.

(In lines 297-299 of page 9)

I cannot find Fig. 3.

We changed Figure 4 to Figure 3. We are sorry we are wrong.

The authors have reported that rho0 cells are more prone to ferroptosis. Please discuss such findings in comparison with the findings in CRR cells.

We compared and discussed ρ0 cells and CRR cells about ferroptosis.
(In lines 299-305 of page 9)

In general, total content of the manuscript is relatively small as a review. If possible, expansion of the content is recommended.

We expand the content and increased total volume.
